# Efficacy of a Positive Psychology Intervention in Enhancing Optimism and Reducing Depression Among University Students: A Quasi-Experimental Study

**DOI:** 10.3390/bs15050571

**Published:** 2025-04-24

**Authors:** Elisenda Tarrats-Pons, Marc Mussons-Torras, Yirsa Jiménez-Pérez

**Affiliations:** 1Department of Economics and Business, Faculty of Business and Communication Studies, University of Vic—Central University of Catalonia, 08500 Vic, Spain; 2Department of Social Psychology and Quantitative Psychology, Faculty of Psychology, University of Barcelona, 08007 Barcelona, Spain

**Keywords:** positive psychology interventions, well-being, depression, optimism, university students

## Abstract

Positive psychology interventions in higher education can be pivotal in reducing depression rates among university students while also enhancing their optimism and well-being. This study aims to examine the effectiveness of implementing a 15-week group intervention on a sample of 194 students who were assigned to either the intervention group (N = 126) or the control group (N = 68). Utilizing a quasi-experimental design with experimental and control conditions, pre- and post-intervention measures were administered to assess depression, optimism, pessimism, and perseverance towards long-term goals. Specifically, the Life Orientation Test-Revised (LOT-R), the Center for Epidemiologic Studies Depression Scale (CES-D), the Attributional Style Questionnaire (ASQ), and the GRIT Scale were employed. The results indicate that students in the experimental group reported significantly higher levels of optimism and reduced depression rates compared to the control group, where no significant differences were observed between pre- and post-intervention outcomes. In conclusion, the implementation of the Hallenges group intervention program may be effective in contributing to the well-being of university students. However, further research is needed to refine and enhance this intervention and to apply it across different university grades and courses.

## 1. Introduction

The 2020 World Happiness Report ([38]) highlights significant international differences in youth happiness perception. In Spain, only 38% of young people self-identify as happy, placing the country in the lower range of the global happiness spectrum, alongside Peru (32%), Chile (35%), Argentina (43%), Hungary (45%), and Mexico (46%) ([29]). In contrast, approximately 87% of the population in the Netherlands reports being happy, ranking the country second worldwide in terms of subjective well-being. In Europe, around 16.6% of young people express dissatisfaction with their lives, though with significant differences: while only 6.7% of 15-year-old adolescents in the Netherlands report low life satisfaction, this figure rises to 25% in the United Kingdom ([60]). These findings underscore the need for evidence-based public policies to enhance youth well-being ([29]).

Youth happiness results from a multidimensional interaction of economic, social, psychological, and cultural factors ([5]; [21]; [61]). Variables such as economic instability, youth unemployment, and job insecurity have been identified as significant stressors contributing to lower subjective well-being among young people in Spain ([57]). Likewise, family support and community cohesion play a fundamental role in promoting happiness and life satisfaction ([36]).

Previous studies have indicated that access to educational opportunities, job prospects, and environmental quality are critical determinants of youth well-being ([6]). A global study involving 10,000 young people aged 16 to 25 revealed that 59% express deep concern about climate change, with nearly half reporting that this environmental anxiety negatively affects their daily lives ([32]).

Differences in happiness levels between Spain and the Netherlands can be attributed to structural factors such as work–life balance, the strength of social support systems, and overall quality of life ([60]). The Organisation for Economic Co-operation and Development (OECD) emphasizes that physical and mental health, the quality of personal relationships, and satisfaction with living conditions are universal determinants of happiness. Furthermore, evidence suggests that investing in mental health services and positive psychology interventions can significantly enhance youth well-being ([39]; [54]; [86]), reducing symptoms of anxiety and depression ([1]; [40]; [80]). The aim of interventions from the framework of positive psychology is not only to promote optimal performance and functioning in people, but also to assess the effectiveness of these interventions. Since [81] ([81]) began to test the effectiveness of a variety of exercises designed from positive psychology to improve psychological well-being and reduce depressive symptoms, the long-term effectiveness of activities related to gratitude, personal strengths, and relaxation in reducing stress, anxiety, and depression has been proven ([31]).

Interventions in the university field such as those of [70] ([70]) and [23] ([23]) point out the benefits of psychoeducational interventions within the framework of positive psychology in self-knowledge, emotional well-being, and academic performance, pointing out that it is necessary to have a different battery of activities and interventions to increase the possibility of identifying procedures adapted to the needs, resources, and preferences of students.

Generation Z, which primarily includes university students and recent graduates, faces significant challenges affecting their mental health ([49]). During the COVID-19 pandemic, the incidence of anxiety, depression, and stress within this group increased markedly, exacerbating economic uncertainty and difficulties in accessing psychological support ([1]; [69]; [72]). The World Health Organization (WHO) defines mental health as a state of well-being in which individuals can cope with daily stressors, work productively, and contribute to their community ([25]). However, globally, more than 13% of adolescents aged 10 to 19 are diagnosed with a mental disorder, and approximately 46,000 adolescents die by suicide each year, equating to one death every 11 min ([91]). Stressors affecting Generation Z include gun violence, political instability, discrimination, and economic pressures ([2]). These elements have contributed to a higher prevalence of mental health issues among this generation compared to previous cohorts. Nevertheless, they also demonstrate a greater willingness to seek professional help, presenting an opportunity to strengthen the provision of mental health services ([2]).

These data indicate a growing dissatisfaction among Spanish youth, necessitating the implementation of education and public health strategies based on positive psychology. Mental health literacy, understood as the ability to recognize disorders, reduce stigma, and seek help effectively, is essential for addressing these challenges ([82]). The disparity in happiness levels between Spain and the Netherlands points to a structural issue requiring sustainable interventions. Early detection of mental health problems in young people can facilitate the development of effective policies and improve their resilience ([58]). In this context, educational institutions play a crucial role in implementing prevention programs and reducing the stigma associated with mental disorders ([13]; [84]).

In response to the growing concern over the deterioration of youth mental health, a positive-psychology-based intervention, termed Hallenges, derived from the concept of Happy Challenge, is proposed. This program, grounded in validated principles of positive psychology ([42]; [50]; [89]), aims to foster optimism ([39]; [74]), resilience ([24]; [39]; [51]), and overall well-being ([39]; [42]; [50]; [54]; [89]), in addition to reducing the incidence of depressive symptoms among young people ([7]; [86]). Hallenges integrates empirically validated methodologies ([46]), such as journaling and gratitude letters, identifying personal strengths, three good things ([39]; [86]), and practicing mindfulness ([42]). Its implementation in a semester-long university course could contribute to greater psychological and academic well-being among students ([19]; [39]; [85]; [86]; [89]).

### 1.1. Positive Psychology Interventions to Promote Optimism and Well-Being

Positive psychology emerged in the late 20th century as a response to the traditional approach focused on mental illness and deficits, instead proposing the scientific study of the most adaptive and health-promoting aspects of human functioning. It was defined by [79] ([79]) as the systematic study of positive experiences, positive individual traits, and institutions that support their development, with the aim of enhancing quality of life and preventing psychopathology. This perspective, in addition to focusing on subjective well-being, incorporates the analysis of human strengths such as resilience, a sense of meaning in life, and optimism ([81]). Complementarily, [83] ([83]) emphasize that positive psychology also addresses human potential, motivations, capabilities, and civic virtues, including social responsibility and community engagement. This approach has contributed to the development of a distinct theoretical and methodological framework, as well as evidence-based interventions aimed at fostering positive emotions, strengthening interpersonal relationships, and cultivating psychological resources for both preventive and therapeutic purposes. Within this framework, optimism is considered one of the most relevant personal strengths for well-being. [74] ([74]) define it as a stable and generalized disposition to expect positive outcomes in the future, thus establishing its dispositional nature. This perspective was further expanded by [76] ([76]), who highlighted its relationship with future expectations and its role as a key variable in psychological adaptation. Numerous studies have supported its association with higher levels of emotional well-being ([52]; [63]; [77]), improved mental health ([81]), and greater life expectancy in clinically vulnerable populations ([74]; [92]). Moreover, optimism has been identified as a moderating factor in the relationship between stressful life events and physical and psychological well-being, acting as a buffer against adversity and a facilitator of positive coping.

One of the key contributions of positive psychology is the development of so-called positive psychology interventions, which comprise a set of theoretically based and empirically validated activities designed to promote the development of beneficial psychological experiences ([39]; [86]) and a full and prosperous existence ([19]; [85]; [89]). In essence, PPIs are essential tools within the field of positive psychology, aimed at promoting well-being, increasing cognitive ([65]) and emotional engagement ([54]), and improving the quality of life in a holistic way ([39]). These interventions aim to strengthen resilience, well-being ([39]; [86]), and the construction of a sense of life purpose, while alleviating depressive symptoms ([11]; [86]). In depression, there is a strong and active network of negative emotions, cognitions, and behaviors, in contrast to a weak network of positive nodes ([28]), suggesting that a positive psychology intervention can contribute to a significant change in the reorganization of these variables ([7]). Furthermore, a meta-analysis of positive psychology interventions concluded that these interventions significantly increase well-being and alleviate depressive symptoms in both clinical and non-clinical samples ([9]; [14]; [85]; [86]). According to [86] ([86]), in addition to improving self-esteem and subjective well-being while reducing self-criticism, especially in highly dependent individuals ([19]), positive psychology interventions (PPIs) are characterized by their multifaceted nature and focus on promoting happiness ([86]) and building character strengths through practices such as expressing gratitude, practicing optimistic thinking ([39]), building on personal strengths, focusing on the positive aspects of life, reliving positive experiences, and socializing ([9]; [11]; [42]). These practices facilitate a more holistic and profound change in participants ([30]; [93]). Among these practices, the consistent expression of gratitude is positively correlated with greater happiness ([39]), greater energy, greater hope for the future, and more positive emotions ([53]). Incorporating diverse practices into a positive psychology intervention increases the likelihood of a positive outcome ([71]) by better accommodating individual differences. Furthermore, group interventions provide a safe and supportive environment that meets relational needs and helps develop greater interpersonal security ([87]). Positive psychology interventions (PPIs) also contribute to the development of optimism ([39]), defined as the generalized expectation of good life outcomes ([74]) and reflecting positive expectations for the near future ([12]). Furthermore, dispositional optimism is positively correlated with decision-making styles in adolescence ([55]).

Several studies have shown that positive psychology interventions (PPIs) generate sustainable impacts by increasing happiness and reducing depressive symptoms, serving as an effective complement to conventional mental health treatment methodologies ([81]). Furthermore, these interventions help people experience positive emotions ([39]), which contributes to an upward spiral that improves resilience and well-being ([24]; [42]) and enables the accumulation of resources to face new challenges ([42]) and improve happiness ([16]). Furthermore, these interventions are low-cost, easy to implement, non-stigmatizing, and free from side effects ([91]). Specialized programs such as the “Penn Resilience Program” and the “Strath Haven Positive Psychology Curriculum” have proven exceptionally effective in fostering resilience and encouraging positive emotions in students, leading to improvements in engagement, behavior, and academic performance ([80]), thus highlighting the transformative potential of PPIs in educational contexts.

University students often face challenges as they transition from adolescence to adulthood, a period requiring significant learning and adaptation ([3]). This period coincides with the maximum chronological age at which certain mental health disorders may emerge ([41]). It represents a peak in academic stress and a significant period of personal adjustment ([90]), where positive psychology interventions integrated into curriculum ([34]) could be highly beneficial and are receiving increasing attention ([44]) given that university student support services are often overburdened and subject to long waiting lists ([68]). Although these interventions are often not applied in higher education ([59]) due to the emphasis on knowledge acquisition and academic performance ([42]), few studies have been applied at the university level. A global systematic review of PPIs identified a total of 27 interventions that included quantitative measures of psychological well-being ([34]). Results reported that 41% of interventions achieved positive effects across all well-being measures, 45% on at least one psychological well-being indicator, 7% showed no effect, and 7% reported negative effects ([34]). In this line, PPIs have been found effective in increasing psychological well-being and reducing depression and anxiety, with benefits lasting between three and six months after the intervention. Moreover, positive psychological well-being enhances students’ confidence in completing their degree programs ([48]) and improves their academic performance ([10]). Among them, a notable study is the 5-week group intervention focusing on positive psychology for university students in Greece ([42]). This intervention led to significant improvements in positive emotions, resilience, and self-esteem, although the impact on self-esteem was less pronounced. In the study by [45] ([45]), a PPI intervention was conducted with university students in the United Arab Emirates, which also concluded that students experienced an increase in positive emotions and greater emotional balance, resulting in more positivity compared to the control group. The intervention by [86] ([86]) demonstrated that the use of technology can facilitate daily reflection exercises, improving stress management in university students. Also, interactivity in mental health interventions improved user adherence and long-term impact. This study is one of the first to examine tangible user interfaces for positive psychology interventions ([86]). [65]’s ([65]) intervention examined the impact on self-efficacy and self-confidence of a University Preparation course among 72 Vietnamese first-year university students (Experimental group n = 50 and Control group n = 22). The results obtained highlight that the experimental group showed a significant increase in self-efficacy and the effects were maintained at the six-week follow-up. In relation to self-confidence, significant improvements were observed in the experimental group in critical thinking, creativity, greater ability to set and achieve learning goals, and greater confidence in the use of digital tools ([65]). [39]’s ([39]) research evaluated the impact of a positive psychology group intervention focused on gratitude on the subjective well-being, optimism, and resilience of undergraduate engineering students (experimental group n = 34 and control group n = 35). The results show significantly higher levels of gratitude, impact on psychological well-being, optimism, improved student engagement, and academic success ([39]). This intervention provides practical recommendations for higher education institutions to implement gratitude-based interventions. Finally, [54]’s ([54]) intervention focused on training 25 English language instruction (EMI) teachers in positive psychology engagement strategies, based on [78]’s ([78]) PERMA model. The IPPs empowered EMI teachers to integrate engagement strategies that improved their students’ learning and psychological well-being ([54]). PPIs can be implemented through diverse approaches ranging from mental health literacy ([43]) and mindfulness practices ([27]) to life skills training ([47]). Regarding student learning, these interventions can promote gratitude expression ([97]), strength recognition and use ([66]), acts of kindness ([17]), active listening, and meditation ([18]).

Despite the growing interest in positive psychology interventions (PPIs) in higher education, the majority of studies identified by [34] ([34]) were conducted in countries such as the United States (k = 11), Australia (k = 6), China (k = 2), and Italy (k = 2), with no empirical evidence reported from Spain. Furthermore, many of these interventions relied on short-term experimental designs or pilot studies, limiting the generalizability of their findings. In this context, the present study offers a novel contribution by implementing a 15-week quasi-experimental intervention within the Spanish university system, including a control group. The results revealed significant increases in optimism and reductions in depression levels, supporting the effectiveness of curriculum-integrated PPIs. This research addresses a critical gap in the international literature by providing context-specific data from Spain and reinforcing the applicability of PPIs within European educational settings.

### 1.2. Aim and Hypotheses

This study aims to examine the effects of a psychoeducational program based on the principles of positive psychology on the psychological variables of optimism and persistence, as well as on the reduction in depressive symptoms in an experimental group. This analysis is grounded in the theoretical and empirical evidence existing in the specialized scientific literature on positive psychology, which posits significant benefits of structured psychoeducational group interventions on these variables.

In this context, our research hypotheses are focused on the following:

**H1.** 
*Participants in the experimental group will show a statistically significant increase in post-intervention optimism levels compared to their pre-intervention levels.*


**H2.** 
*Participants in the experimental group will achieve a statistically significant decrease in post-intervention depression levels compared to initial measurements.*


**H3.** 
*Participants in the experimental group will exhibit a statistically significant increase in long-term goal persistence following the intervention.*


**H4.** 
*There will be no statistically significant changes in the measurements of optimism, depression, and long-term goal persistence among students in the control group throughout the intervention (pre- and post-program evaluation).*


In response to this emerging challenge, the proposal of a targeted positive psychology program, Hallenges offers a promising avenue to address mental health issues prevalent among university students. Anchored in the core principles of positive psychology, this intervention seeks to cultivate resilience, improve well-being, and decrease the prevalence of depressive symptoms within this cohort. By integrating evidence-based interventions such as gratitude journaling, strength identification, and mindfulness practices, the program aims to foster a strong psychological framework among young adults, allowing them to thrive academically and personally. The hypothesis driving this initiative postulates that, through the systematic application of the Hallenges program, a measurable improvement in the emotional well-being of university students can be achieved, which could mitigate the incidence of depression and increase their general happiness ([79]).

## 2. Materials and Methods

### 2.1. Participants

A total of 194 first-year students, comprising 76 men and 118 women, were selected from the undergraduate programs in criminology and psychology at ESERP, a center affiliated with the University of Vic—Central University of Catalonia (UVic-UCC) and the University of Barcelona (UB). Prior to the commencement of the study, participants were provided with detailed information about the research objectives, and their informed consent was obtained, ensuring their full understanding of the implications of their involvement. To maintain data confidentiality, students completed the measurement instruments through the online platform https://www.hallenges.com/, which is configured to ensure the confidentiality of responses. It is emphasized that participation was voluntary, and no financial incentives were offered for their collaboration. Additionally, the study design and procedures were reviewed and received favorable approval from the Ethics Committee of UVic-UCC, confirming adherence to ethical research standards as well as the ethical principles of the Helsinki Declaration. Each subject where two parallel groups existed, one was randomly assigned as the experimental group and the other as the control group, ensuring that both groups followed the same course content and teaching staff. The inclusion criteria required that students had attended the full 15-week intervention period and completed both the pre-test and post-test questionnaires. Students who did not complete these two measurements were excluded from the final sample.

### 2.2. Procedure

In the initial phase, both the control and experimental groups completed a battery of psychometric questionnaires in person, which included the Life Orientation Test-Revised (LOT-R) ([75]), the Center for Epidemiologic Studies Depression Scale (CES-D) ([67]), the Attributional Style Questionnaire (ASQ) ([64]), and the GRIT test ([22]) for measuring persistence and sustained effort.

The group intervention focused on positive psychology was structured within the regular curriculum, allocating 10 min per week during a semester in an undergraduate course. The purpose was to understand and apply fundamental concepts of positive psychology through practical activities tailored to the students’ needs. Selected activities included identifying and recording ‘three good things’, writing gratitude letters, using personal strengths, practicing *savoring*, and creating personalized action plans.

In particular, for the experimental group, each session began with a didactic presentation by the instructors, who were previously trained in the program’s implementation. During these presentations, theoretical content and the practical activities corresponding to the session were introduced. Subsequently, students applied these activities in their daily routines, recording their experiences in specific provided forms. Additionally, the course’s online platform served as a repository for materials and a space for the submission of weekly assignments. During the sessions, dialogue among participants was encouraged regarding their reflections and experiences derived from the intervention.

It is important to note that the control group did not receive any intervention during this process. At the end of the intervention period, participants in both groups were re-evaluated using the same initial psychometric instruments to examine the effects of the intervention.

The quasi-experimental design approach used in this research allows for a comprehensive evaluation that integrates both the evolution of participants throughout the intervention and comparisons between the experimental and control groups, providing a deeper understanding of the intervention’s effects. This design is particularly valuable for discerning individual variations in response to the intervention, as well as for contrasting these changes with a non-exposed group, thereby strengthening the robustness and internal validity of the findings.

Each student was assigned a unique identifier to ensure traceability throughout the study in both the control and experimental groups. In the control group, evaluations were conducted at two different times, with a fixed interval between them, without introducing any type of positive psychology intervention during this period. The procedure for the experimental group followed a similar scheme; however, between the two evaluations, a 15-week positive psychology intervention called Hallenges was administered (the theme of each session and the expected outcomes are detailed in Table 1). The central hypothesis of the study posits that this intervention can induce significant variations in the psychological metrics evaluated, specifically regarding optimism, perseverance, subjective happiness, and depressive symptoms.

The comparative methodology will involve a cross-analysis of the metrics between the control and experimental groups, using unique identifiers to ensure precise longitudinal evaluation of the participants. It is anticipated that the control group will not show statistically significant differences between the pre-established pre- and post-measurements, given that the intervention was not part of their protocol. This will contrast with the expected data from the experimental group, where the positive psychology intervention is anticipated to modify the study variables.

The control group consists of 68 students, with a gender distribution of 79.4% female and 20.6% male. In terms of marital status, 60.3% of the participants identify as single or in relationships of less than one year, while 39.7% report being in relationships of more than one year. The academic composition of the group includes 92.6% of students enrolled in the criminology program and 7.4% in psychology, from the University of Vic—Central University of Catalonia (UVIC-UCC) and the University of Barcelona (UB), respectively.

On the other hand, the experimental group, consisting of 126 students who participated in the positive psychology intervention Hallenges, has a gender distribution of 82.5% female and 17.5% male. Regarding marital status, 68.7% are classified as single or in relationships of less than one year, and 31.3% in relationships of more than one year. The disciplinary distribution for this group is 56.3% criminology students and 43.7% psychology students, with an academic background identical to that of the control group.

It is noteworthy that a non-probabilistic intentional sampling method was used for the selection of the cohorts. The units of analysis consisted of specific undergraduate classes at the university, selected for their relevance to the constructs of interest. Once established, they were deliberately assigned to the control and experimental groups to evaluate the impact of the intervention. This strategic assignment allowed for the control of contextual variables inherent to the academic environment and facilitated the administration of the evaluation measures. (See Table 2).

### 2.3. Measures

In the present study, measurements of key psychometric constructs were conducted using validated instruments. Optimism was measured using the Life Orientation Test-Revised (LOT-R) ([75]), with scores ranging from 0, indicating the absence of optimism, to a maximum of over 24, representing a high level of optimism. The prevalence of depressive symptoms was determined using the Center for Epidemiologic Studies Depression Scale (CES-D) ([67]), which ranges from 0 to 60, with higher scores reflecting greater severity of depressive symptoms. Persistence and sustained effort were evaluated using the GRIT scale ([22]), where the range extends from 0 to 5, with 5 indicating the highest possible persistence. Additionally, Seligman’s Attributional Style Questionnaire (ASQ) ([64]) was applied for a multifaceted assessment of optimism, using the PvG, PvB, PmG, and PmB subscales, each with a scoring range from 0 to 8, where lower scores indicate lower optimism. Self-esteem was quantified using the PsB and PsG scales, both with a range from 0 (lowest self-esteem) to 8 (highest self-esteem). Finally, hope was measured by the HoB scale, which ranges from 0 (no hope) to 16 (maximum level of hope), thus providing a detailed profile of this psychological construct.

### 2.4. Statistical Analysis

The statistical analysis of the data began by verifying linearity using scatter plots. Subsequently, the normality of the distributions was examined through histograms, Q-Q plots, goodness-of-fit tests such as Kolmogorov–Smirnov and Shapiro–Wilk, and analyses of skewness and kurtosis, where no significant deviations were identified.

To compare the evaluated psychological factors—optimism, depression, and persistence—means and standard deviations were calculated, and Student’s *t*-tests for independent samples were performed at two points, before and after the intervention. Additionally, a multivariate analysis of variance (MANOVA) was conducted to contrast positive and negative emotions, resilience, subjective happiness, optimism, and self-esteem between the conditions of positive intervention and the absence of it before the program.

Intragroup comparisons were conducted using the paired-samples *t*-test, evaluating differences at pre- and post-intervention points. Finally, mixed ANOVAs were applied to determine the effect of the intervention, considering the experimental condition as a between-subjects variable and time (pre–post intervention) as a within-subjects factor.

Data processing was conducted using the statistical software SPSS, version 21, ensuring the traceability and confidentiality of the data through unique codes assigned to each student in both the control and experimental groups. The significance level adopted for all statistical tests was 0.05, according to standard conventions in psychological research.

This statistical approach was selected to maximize the understanding of the dynamics and effects of the Hallenges training program on the psychological variables of interest, ensuring rigor and precision in the interpretation of the obtained results.

## 3. Results

Initially, to evaluate the effectiveness of the program, we compared pre- and post-intervention changes in various psychological measures in a sample of 194 university students (N = 126 experimental group and N = 68 control group). The results of the non-parametric Mann–Whitney U and Wilcoxon W tests did not reveal significant differences in the control group, confirming the stability in levels of optimism, depression, persistence, and self-esteem in the absence of the program.

Consequently, we validated Hypothesis 4. All null hypotheses are accepted, as observed in the non-parametric contrast Table 3, Table 4 and Table 5, with a significance level of 0.05. Therefore, all students who completed the questionnaire before and after the duration of the Hallenges program, and did not undergo the intervention, show no differences in these factors and no improvement in any indicator of optimism, self-esteem, hope, or persistence. To test the hypotheses after the intervention, two group comparisons were conducted: (1) between the post-intervention control group and the post-intervention experimental group, and (2) between the pre- and post-intervention scores of the experimental group. In both analyses, the null hypothesis was rejected for the CESS-D and LOT-R variables (see Table 6 and Table 7).

### 3.1. Measures of Optimism (LOT-R)

An analysis of variance (ANOVA) revealed significant improvements in optimism levels as measured by the LOT-R questionnaire (Table 8). The Student *t*-test (Table 9) rejects the null hypothesis (significance 0.032). The post-intervention results showed a higher median compared to the pre-intervention, with a significant increase of 1 point in the mean (*p* < 0.048). This validates Hypothesis 1, indicating that students who participated in the program experienced an increase in their level of optimism, rising from a mean of 15.54 to 16.54 points, and rejects the null hypothesis that there are no significant differences between the measurements before and after the intervention.

As can be seen in the box plot (Figure 1), the median of the scores in the post sample is higher than that of the pre sample (17 vs. 16). Additionally, the 75th percentile in the post sample is above that of the pre sample (20 vs. 19), and the 90th percentile also shows an increase (22 vs. 21).

The 95% confidence interval for the pre-intervention scores ranges from 14.77 to 16.30, while the post-intervention scores range from 15.79 to 17.28, indicating a significant improvement in optimism levels following the intervention.

The box plot (Figure 1) illustrates an upward shift in the distribution of scores, visually confirming the improvements observed in the statistical analysis.

### 3.2. Depression Measures (CES-D)

Regarding the CES-D factor, the null hypothesis is rejected using the Student *t*-test (Table 9), significance < 0.001. A significant average decrease of 4.7 points in depressive symptoms was observed, from a pre-intervention mean of 21.51 to a post-intervention mean of 16.80 (*p* < 0.001). This significant reduction in depressive symptoms is evident in the box plot (Figure 2), where the median decreases from 19 to 15.5, and the 75th percentile drops from 29 to 24.

The confidence interval also shows a notable decrease, shifting from a range of 19.69–23.34 in the pre-intervention to 14.73–18.86 in the post-intervention. This change indicates that the minimum of the pre-intervention range (19.69) is higher than the maximum of the post-intervention range (18.86), highlighting the effectiveness of the program in reducing depressive symptoms. These results validate Hypothesis 2 of our study, demonstrating that the intervention produces a significant improvement in the emotional state of the participants.

### 3.3. Persistence Measures (GRIT)

Regarding persistence (GRIT), although there were no statistically significant changes, the box plots suggest a trend towards greater persistence in the group that completed the Hallenges program. The Student *t*-test (Table 9) does not reject the null hypothesis (significance 0.404). Consequently, this refutes Hypothesis 3.

### 3.4. Correlation Analysis of Psychological Measures

Finally, the Spearman correlation matrix (Table 10) revealed significant associations, particularly between measures of optimism and hope (PmB and HoB, PvB and HoB). There was also a notable inverse correlation between levels of depression and optimism (CES-D and LOT-R). These correlations support the theory that hope and optimism are interrelated and both inversely associated with depression. Additionally, strong correlations were observed between self-esteem and hope (PsB and HoB, PsG and HoB) and between persistence (GRIT) and positive attributional style (PvG). These findings underscore the interconnected nature of positive psychological constructs and the role of optimism and hope in mitigating depressive symptoms and enhancing overall well-being. The significant negative correlations between depressive symptoms (CESS-D) and both self-esteem (PsB, PsG) and optimism (LOT-R) further validate the effectiveness of positive psychology interventions in reducing depressive symptoms and improving mental health outcomes. It is worth noting that the correlation between LOT-R and CESS-D also remains strong when calculated between the post-intervention control group and the post-intervention experimental group. In this case, a Spearman’s rho of ρ = –0.624 was obtained, with a significance level of *p* < 0.001.

## 4. Discussion

The results of this study demonstrate that the Hallenges intervention, structured across 15 sessions and embedded within the university curriculum, produced significant improvements in variables such as optimism and reduced depressive symptoms. These findings align with a robust body of literature supporting the effectiveness of positive psychology interventions (PPIs) in promoting psychological well-being among university students ([18]; [20]; [26]; [56]; [73]; [88]; [95]), in both clinical and non-clinical populations ([9]; [39]; [85]; [86]). Furthermore, these results respond to the urgent need for innovative proposals to address the mental health crisis in higher education, as emphasized by [34] ([34]) in their systematic review of 27 interventions. They also support recent findings by [54] ([54]) and [65] ([65]), who highlight the positive effects of PPIs on self-esteem, optimism, and depression reduction.

A key component that may have contributed to these outcomes was the integration of the intervention into a mandatory course. Each session concluded with 15 min dedicated to delivering evidence-based content and reflective activities aimed at increasing students’ positive emotions, cognitions, and behaviors ([9]). This curricular approach facilitated the transfer of psychological skills to everyday life and enabled a progressive development of self-awareness and personal growth throughout the 15-week period, consistent with the average duration of effective university-based interventions ([34]). The multicomponent design of the program, which included various empirically supported techniques, also contributed to its effectiveness. Specifically, the intervention incorporated life skills training ([47]), expressions of gratitude ([97]), use of personal strengths ([66]), savoring ([15]; [26]; [95]), positive visualization ([95]), meaning in life ([18]), and acts of kindness ([17]). These practices have been shown to enhance psychological well-being and reduce depressive symptoms ([8]; [30]; [39]; [88]), while also promoting sustained engagement in the intervention ([9]).

The group format of the program further contributed to its success. As noted in the literature, group-based interventions provide a psychologically safe and supportive environment that fosters interpersonal connection and emotional security, especially among individuals with higher emotional dependency ([87]). In this context, gratitude-based activities encouraged prosocial behavior and acts of kindness toward others, reinforcing their effectiveness as core components of PPIs ([30]).

However, the intervention did not yield significant changes in *grit* (long-term goal persistence). This finding is consistent with prior research suggesting that grit may function more as a stable trait than a modifiable state, and thus may be less responsive to short-term interventions ([37]). Additional studies have indicated that the development of grit relies on specific factors such as metacognition, hope, and life purpose ([88]), which, although present in many PPIs, were not explicitly addressed in this program. Similarly, research involving specific populations—such as Latino students—has found that grit is more strongly associated with hope and meaning in life than with direct intervention exposure ([33]; [96]).

Regarding the internal validity of the findings, it is essential to acknowledge the potential influence of uncontrolled confounding variables. Although academic homogeneity and balanced group assignment were maintained, it was not possible to fully control for individual variability such as prior life experiences, baseline motivation, or emotional states. These factors may have influenced both the degree of engagement and responsiveness to the intervention. Future studies are encouraged to incorporate moderating variables such as personality traits, perceived stress levels, and qualitative assessments of participant experiences to better understand the underlying mechanisms of change and the contextual factors that may mediate intervention effectiveness. Moreover, the influence of broader external factors should be considered. For instance, participants’ prior exposure to critical life events—such as the COVID-19 pandemic during early adulthood—may have impacted their emotional regulation or perseverance. Additional psychosocial variables, including family environment, social support networks, or academic pressure from concurrent coursework, were not measured in the current study. The exclusion of these elements constitutes a key limitation and underscores the importance of designing future longitudinal research that includes these variables as potential moderators. Integrating such dimensions would not only enable a more comprehensive evaluation of intervention outcomes, but also facilitate the adaptation of PPI content to better meet the specific needs of diverse student populations.

## 5. Limitations and Future Research

Although the results obtained from the intervention are interesting, it is important to highlight several limitations of the study. First, although the sample size is large, it only includes students from two university degree programs, which limits the generalizability of the results to other groups and academic disciplines ([54]; [65]; [86]). Additionally, the design used was quasi-experimental and not randomized, introducing potential biases due to the lack of randomness in sample selection and the assignment of entire classes to control or experimental groups without randomization ([39]). This methodology may increase the likelihood that uncontrolled variables could influence the results. To strengthen the internal validity and generalizability of the findings, it is preferable to use probabilistic sampling with random assignment in quantitative studies. Another additional limitation is the exclusive use of self-reported measures to assess variables such as depression, optimism, and perseverance, which may introduce social desirability bias, as participants might have provided responses they perceived as socially acceptable rather than entirely truthful ([94]). Furthermore, subjective biases related to limited or inaccurate introspective ability should be considered, as these could hinder participants’ capacity to accurately evaluate their emotional states and personal traits ([62]). These limitations may have affected both the precision of the measurements and the interpretation of the observed effects. However, it is worth noting that the existence of an experimental group and a control group in this intervention contributes to greater internal validity of the study ([39]; [65]).

For future research, it is recommended to explore various avenues to expand and deepen the current findings. First, it would be valuable to broaden the sample to include students from different academic programs and disciplines ([39]; [65]). This would allow for an assessment of the generalizability of the results to a wider and more diverse student population. Additionally, it is suggested to conduct a longitudinal analysis to examine the long-term impact of positive psychology interventions (PPIs) ([39]; [65]; [86]). According to previous studies there is not yet sufficient evidence to support the long-term beneficial effects of PPIs beyond four months after their implementation ([34]). Therefore, future studies should include long-term follow-up measures to help estimate how long the effects of these interventions are sustained. This type of design could also facilitate the observation of temporal variations based on contextual or individual variables. Moreover, it is crucial to explore cultural adaptations that would allow the intervention to be tested and validated in more diverse educational and sociocultural contexts. This would help to determine the effectiveness of PPIs across different cultural settings, ensuring cultural relevance and sensitivity of the strategies implemented. This approach would enable the observation of how the effects of the interventions are sustained over time and whether changes in well-being and other psychological indicators occur at different stages of the educational cycle. Including personality variables in the study design could also provide valuable insights into which personal characteristics modulate the effectiveness of PPIs. Assessing whether these interventions are more suitable for individuals with certain personality traits, such as resilience, self-criticism, or dependency, could help to tailor interventions and maximize their effectiveness. Finally, the incorporation of gamified technological applications as support for PPIs should be considered ([86]). This approach could enhance participant engagement and adherence, especially among the millennial generation, who exhibit a greater affinity for technology and digital platforms. Evaluating the impact of these technological tools on intervention outcomes could offer a promising avenue to increase the efficacy and reach of PPIs. These proposals would not only expand the current knowledge on the effectiveness of PPIs but also contribute to the development of more adapted and personalized interventions, enhancing their applicability in various educational contexts.

## 6. Conclusions

In a BANI environment (Brittle, Anxious, Non-linear, and Incomprehensible), the psychological well-being of university students poses an increasing challenge for educators and academic institutions ([4]). In this context, implementing positive psychology programs in higher education emerges as a promising strategy to promote mental and emotional health among students. The Hallenges intervention, grounded in validated principles of positive psychology and integrated into regular coursework, proved effective in increasing optimism and reducing depressive symptoms, thereby replicating outcomes observed in previous research ([39]; [54]; [86]).

Key components that likely contributed to the program’s effectiveness include the use of empirically supported activities—such as gratitude letter writing, personal strength identification, and savoring practices—as well as its group-based format, which provided a safe environment that encouraged emotional engagement and active participation ([24]; [87]). The observed increase in optimism and the significant reduction in depressive symptoms reinforce the value of positive psychology interventions (PPIs) as accessible and sustainable tools for educational settings.

However, the intervention did not produce significant changes in the measure of grit (persistence), aligning with previous findings that suggest this construct may be relatively stable and less responsive to short-term interventions ([37]). This result highlights the need for longer or more targeted interventions when aiming to foster long-term goal persistence.

From a practical perspective, these findings support the integration of positive psychology content into university curricula to promote students’ holistic development. Future studies are encouraged to expand the sample to include students from other academic disciplines and to explore the long-term impact of such interventions, as well as the moderating role of individual characteristics such as resilience, self-criticism, or emotional dependency ([19]). Additionally, incorporating technological tools may enhance adherence and effectiveness, particularly among younger generations ([86]).

In conclusion, the Hallenges intervention provides empirical evidence of the value of PPIs in university contexts. Its systematic integration into academic programs can significantly contribute to improving emotional well-being, preventing depressive symptoms, and strengthening personal competencies essential for navigating academic and professional challenges.

## Figures and Tables

**Figure 1 behavsci-15-00571-f001:**
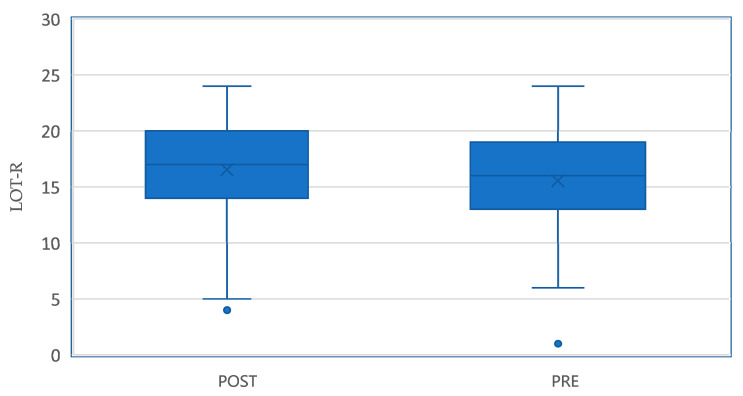
Comparison of pre- and post-intervention optimism scores (LOT-R).

**Figure 2 behavsci-15-00571-f002:**
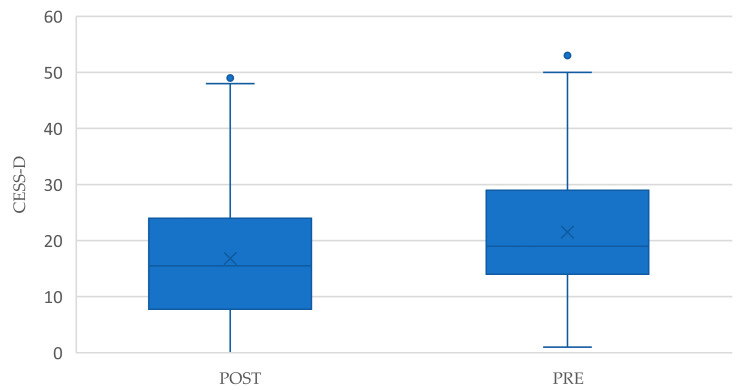
Comparison of pre- and post-intervention optimism scores (CESS-D).

**Table 1 behavsci-15-00571-t001:** Benefits of positive psychology interventions (PPIs) in higher education.

Benefit of PPIs	Author(s) Year	Result/Implication
Improves adaptation and resilience during peak academic stress	([3]; [41]; [90])	Highlights the transition period as critical for mental health; PPIs may ease personal adjustment
Reduces pressure on overwhelmed university support services	([35]; [44])	Curriculum-integrated PPIs can serve as scalable mental health support
Low implementation in higher education despite potential	([42])	Few PPI studies are applied at the university level due to academic focus on performance
Quantified impact across psychological well-being	([34])	41% showed full positive effects; 45% partial; 7% no effect; 7% negative effect
Sustained well-being, reduction in depression and anxiety	([34])	Benefits sustained between 3 and 6 months after intervention
Boosts academic confidence and performance	([10]; [48])	Increased degree completion confidence and improved academic results
Improves emotions, resilience, and self-esteem (Greece)	([42])	Group intervention led to increased well-being, slight effect on self-esteem
Enhances emotional balance and positivity (UAE)	([45])	PPI improved emotional regulation vs. control group
Tech-enabled reflection improves stress management	([86])	Daily digital exercises increased adherence and long-term effects
Increases self-efficacy, critical thinking, creativity	([65])	Vietnamese students showed significant growth and follow-up retention
Gratitude-focused PPI improves engagement and success	([39])	Increased gratitude, optimism, student engagement, and academic success
PERMA-based strategies improve student well-being	([54])	Trained EMI teachers integrated PPIs improving learning and mental health
Diverse delivery methods increase flexibility	([27]; [43]; [47])	PPIs range from literacy to mindfulness and life skills training
Promotes strengths, gratitude, kindness, and mindfulness	([17]; [18]; [66]; [97])	Key activities contribute to positive emotions and personal growth

**Table 2 behavsci-15-00571-t002:** Description of the sessions in the positive psychology intervention program (15 sessions).

Session No.	Theme	Expected Outcome
1	Introduction to the Hallenges program and baseline assessment	Set personal expectations, explore the concept of happiness, and complete baseline self-assessments of optimism/pessimism.
2	Locus of control	Understand and reflect on the internal and external locus of control through personal experiences.
3	Values	Distinguish between values and beliefs, identify personal values, and design an action plan to apply them.
4	Positive psychology and self-leadership	Recognize learned helplessness and its personal impact. Identify depressive symptoms through a self-assessment test.
5	ABC-model	Familiarize students with the model explaining how individuals may respond differently to the same event depending on their underlying beliefs.
6	Optimistic and pessimistic thinking patterns	Identify thinking patterns and how they relate to a more optimistic or pessimistic outlook on life
7	PERMA model of well-being	Identify key components of personal well-being based on the PERMA model and areas for potential improvement.
8	Rational emotive approach	Identify irrational beliefs and learn to replace them with more rational and positive thoughts.
9	Character strengths	Identify personal character strengths and apply them intentionally to enhance well-being, engagement, and life meaning.
10	Passion, perseverance, and gratitude	Set meaningful goals, develop perseverance, and cultivate gratitude to foster resilience and well-being.
11	Time management	Analyze how time is spent on activities that bring the most value: relationships, meaningful work, and personal growth.
12	Coaching and goal setting	Define academic and personal goals and create a structured plan to achieve them.
13	Global well-being	Understand global well-being as an indicator of mental health and reflect on how to positively influence one’s environment.
14	Process integration	Share personal experiences, identify the most effective practices, and consolidate the improvement process.
15	Development of the Hallenges portfolio	Reflect on personal and professional growth by creating a portfolio, and provide feedback on the overall process.

**Table 3 behavsci-15-00571-t003:** Mann–Whitney U test statistics comparing LOT-R, CESS-D, GRIT, and PvG.

N = 136	LOT-R	CESS-D	GRIT	PvG
Mann–Whitney U	2388.500	2383.500	2030.000	2647.500
Wilcoxon W	4734.500	4729.500	4376.000	4993.500
Test Statistic	2388.500	2383.500	2030.000	2647.500
Standard Error	229.171	229.640	228.605	224.675
Standardized Test Statistic	0.334	0.311	−1.234	1.493
Asymptotic Sig. (2-sided test)	0.739	0.756	0.217	0.135

**Table 4 behavsci-15-00571-t004:** Mann–Whitney U test statistics comparing PsB, HoB, PvB, and PmG.

N = 136	PsB	HoB	PvB	PmG
Mann–Whitney U	2255.500	2557.000	2660.000	2239.000
Wilcoxon W	4601.500	4903.000	5006.000	4585.000
Test Statistic	2255.500	2557.000	2660.000	2239.000
Standard Error	226.044	227.001	223.795	223.224
Standardized Test Statistic	−0.250	1.079	1.555	−0.327
Asymptotic Sig. (2-sided test)	0.803	0.280	0.120	0.744

**Table 5 behavsci-15-00571-t005:** Mann–Whitney U test statistics comparing PmB and PsG.

N = 136	PmB	PsG
Mann–Whitney U	2378.000	2330.000
Wilcoxon W	4724.000	4676.000
Test Statistic	2378.000	2330.000
Standard Error	222.786	223.275
Standardized Test Statistic	0.296	0.081
Asymptotic Sig. (2-sided test)	0.767	0.936

**Table 6 behavsci-15-00571-t006:** Mann–Whitney U test statistics comparing Grit, Cess-D, and Lot-R from control sample post versus experimental sample post.

Independent-Samples Mann–Whitney U Test Summary	GRIT	CESS-D	LOT-R
Total N	194	194	194
Mann–Whitney U	4456	3153.5	5465
Wilcoxon W	12,457	11,154.5	13,466
Test Statistic	4456	3153.5	5465
Standard Error	371.601	372.954	372.24
Standardized Test Statistic	0.463	−3.031	3.173
Asymptotic Sig. (2-sided test)	0.643	0.002	0.002

**Table 7 behavsci-15-00571-t007:** Mann–Whitney U test statistics comparing Grit, Cess-D, and Lot-R from experimental sample pre versus experimental sample post.

Independent-Samples Mann–Whitney U Test Summary	GRIT	CESS-D	LOT-R
Total N	252	252	252
Mann–Whitney U	7107.5	10,007	6796
Wilcoxon W	15,108.5	18,008	14,797
Test Statistic	7107.5	10,007	6796
Standard Error	575.487	578.278	576.901
Standardized Test Statistic	−1.443	3.578	−1.98
Asymptotic Sig. (2-sided test)	0.149	<0.001	0.048

**Table 8 behavsci-15-00571-t008:** ANOVA for Lot-R.

LOT-R	Sum of Squares	df	Mean Square	F	Sig.
Between Groups	2121.49	47	45.138	3.651	<0.001
Within Groups	2522.113	204	12.363		
Total	4643.603	251			

**Table 9 behavsci-15-00571-t009:** Student *t*-test for equality means.

	t	df	Significance	Mean Diff	Std. Error Diff	95% Conf Interval of the Diff	
			One-Sided *p*			Lower	Upper
GRIT	−0.244	250	0.404	−0.37302	1.53046	−3.38726	2.64123
CESS-D	3.386	250	<0.001	4.71429	1.39244	1.97188	7.45669
LOT-R	−1.854	250	0.032	−1	0.53929	−2.06213	0.06213

**Table 10 behavsci-15-00571-t010:** Spearman’s correlation coefficients among psychological constructs and well-being indicators.

		PsG	PmB	PmG	PvB	HoB	PsB	PvG	GRIT	CESS-D	LOT-R
PsG	Correlation Coefficient	1.000	−0.089	0.163 **	−0.104	0.055	−0.201 **	0.192 **	−0.018	−0.239 **	0.174 **
	Sig. (2-tailed)		0.158	0.009	0.101	0.385	0.001	0.002	0.771	<0.001	0.006
	N	252	252	252	252	252	252	252	252	252	252
PmB	Correlation Coefficient	−0.089	1.000	−0.019	0.148 *	0.644 **	0.165 **	−0.099	0.092	0.140 *	−0.115
	Sig. (2-tailed)	0.158		0.764	0.018	<0.001	0.009	0.117	0.146	0.026	0.067
	N	252	252	252	252	252	252	252	252	252	252
PmG	Correlation Coefficient	0.163 **	−0.019	1.000	−0.118	0.011	−0.098	0.092	0.025	−0.167 **	0.024
	Sig. (2-tailed)	0.009	0.764		0.062	0.864	0.121	0.146	0.693	0.008	0.703
	N	252	252	252	252	252	252	252	252	252	252
PvB	Correlation Coefficient	−0.104	0.148 *	−0.118	1.000	0.579 **	0.016	−0.058	−0.047	−0.025	0.058
	Sig. (2-tailed)	0.101	0.018	0.062		<0.001	0.804	0.359	0.454	0.697	0.360
	N	252	252	252	252	252	252	252	252	252	252
HoB	Correlation Coefficient	0.055	0.644 **	0.011	0.579 **	1.000	−0.079	0.083	0.152 *	0.029	−0.061
	Sig. (2-tailed)	0.385	<0.001	0.864	<0.001		0.209	0.191	0.016	0.650	0.333
	N	252	252	252	252	252	252	252	252	252	252
PsB	Correlation Coefficient	−0.201 **	0.165 **	−0.098	0.016	−0.079	1.000	−0.242 **	−0.073	0.040	−0.029
	Sig. (2-tailed)	0.001	0.009	0.121	0.804	0.209		<0.001	0.248	0.527	0.649
	N	252	252	252	252	252	252	252	252	252	252
PvG	Correlation Coefficient	0.192 **	−0.099	0.092	−0.058	0.083	−0.242 **	1.000	0.135 *	−0.104	0.092
	Sig. (2-tailed)	0.002	0.117	0.146	0.359	0.191	<0.001		0.032	0.099	0.144
	N	252	252	252	252	252	252	252	252	252	252
GRIT	Correlation Coefficient	−0.018	0.092	0.025	−0.047	0.152 *	−0.073	0.135 *	1.000	0.074	0.025
	Sig. (2-tailed)	0.771	0.146	0.693	0.454	0.016	0.248	0.032		0.243	0.694
	N	252	252	252	252	252	252	252	252	252	252
CESS-D	Correlation Coefficient	−0.239 **	0.140 *	−0.167 **	−0.025	0.029	0.040	−0.104	0.074	1.000	−0.550 **
	Sig. (2-tailed)	<0.001	0.026	0.008	0.697	0.650	0.527	0.099	0.243		<0.001
	N	252	252	252	252	252	252	252	252	252	252
LOT-R	Correlation Coefficient	0.174 **	−0.115	0.024	0.058	−0.061	−0.029	0.092	0.025	−0.550 **	1.000
	Sig. (2-tailed)	0.006	0.067	0.703	0.360	0.333	0.649	0.144	0.694	<0.001	
	N	252	252	252	252	252	252	252	252	252	252

**. Correlation is significant at the 0.01 level (2-tailed). *. Correlation is significant at the 0.05 level (2-tailed).

## Data Availability

The original data presented in the study and the used questionnaire are openly available in The Open Science Framework repository https://doi.org/10.34810/data1790.

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
