# Peer review of "Efficacy of a Positive Psychology Intervention in Enhancing Optimism and Reducing Depression Among University Students: A Quasi-Experimental Study"

_behavsci, 2025, doi:10.3390/bs15050571_

Round 1

Reviewer 1 Report

Comments and Suggestions for Authors

The topic of this research is very important, thanks for the opportunity to review this study. Please see my comments below,

  1. Introduction: psychoeducational program has been introduced in the aim and hypothesis section, but it would be beneficial to introduce the concept and provide literature for the efficacy of psychoeducational programs.
  2. Procedure: There should be a section introducing and explaining the intervention conducted in the study in details for a practical implication. I understand the length limitation of the journal, the authors could also include this as a supplementary document.
  3. Analysis: I wonder why the authors did not propose and test the differences between the experimental and control groups? I would like to see some tests, such as anova between experimental and control conditions in post test among all the outcomes.
  4. Results: for table 4, not sure how important to keep the sig (2-tailed) and N as long as the significance are marked and N is mentioned below the table.
  5. Discussion: this section needs some in-depth discussion, what components worked well in the intervention in these significant results? Additionally, Persistence Measure was not significant, what is the explanation of this insignificance?

Author Response

 Please find attached a word document with the completed revision.

Reviewer 2 Report

Comments and Suggestions for Authors

Dear Authors,

I would like to express my gratitude for the opportunity to review your manuscript. I approached the article with great enthusiasm and believe that it offers valuable insights into the Efficacy of a Positive Psychology Intervention in Enhancing Optimism and Reducing Depression Among University Stu-dents: A Quasi-Experimental Study. The manuscript is written in a very clear and easy to follow language. The authors report an abundance of results in a lot of detail. I highly commend the authors for their due diligence!  Please find below the suggestions and recommendations for improving your manuscript:

  1. Title of the manuscript
  • The title effectively reflects the content of the manuscript, and the topic itself is original and meaningful to the study.
  1. Over Quality of the articles

This study fills a gap in the literature by exploring the Efficacy of a Positive Psychology Intervention in Enhancing Optimism and Reducing Depression Among University Stu-dents, providing valuable insights for further research and intervention development and the study employed currently recommended methods. Besides, the measures of optimism (LOT-R), depression (CES-D), and persistence (GRIT) are thoroughly examined and explained in detail. However, the analysis lacks important information regarding divergent, discriminant, convergent, and construct validities, as well as reliability metrics such as composite reliability and Cronbach's alpha.

I highly recommend including these validity assessments and reliability statistics to strengthen the study. Additionally, measurement invariance across groups should be addressed. This information would enhance the rigor of the preliminary sections and provide a more comprehensive understanding of the psychometric properties of the measures used. Thank you for considering these suggestions.

  1. Originality
  • The research is original. However, it needs recent literature to make the argument more scientific specially the positive psychology areas such as mindfulness, emotional intelligence, psychological capital (efficacy, optimism, resilience and hope), PERMA profiler, gratitude etc. I am eager to see such components in your revised version.
  1. Abstract
  • The abstract section is one of the strengths of the authors' work, as it effectively summarizes the main objectives, methods, results, and conclusions of the study.
  1. Introduction/Literature
  • Optimism as a positive psychological component derived from positive psychology. Make a distinction between positive psychology and optimism, and provide rich evidence.
  • The introduction provides a solid foundation, but consider refining your thesis statement to clearly outline the specific aims and significance of your study. This will help set a clear expectation for the reader.
  • You might want to explicitly state the gap in the literature that your study addresses. Highlighting this will strengthen the rationale for your research.
  • While you cite several relevant studies, a deeper analysis of existing interventions in positive psychology specific to university settings would enhance the context. I recommend that the authors cite different aspects of positive psychology in higher education to provide solid evidence, such as the PERMA profiler, psychological capital, emotional intelligence, optimism, and gratitude in higher education settings. This will enhance your research more meaningful and evidence based.
  • Consider summarizing the main findings of key studies in a table format, which can provide a quick reference for readers and highlight the effectiveness of different interventions. Because the issue of positive psychology is timely and new.
  1. Methodology section
  • The description of your sample size (194 students) is commendable, but please provide more detail about participant recruitment and any inclusion/exclusion criteria. This information is crucial for assessing the external validity of your findings.
  • Additionally, clarify the procedures for the intervention, including how sessions were structured and the specific activities included each week. It would also be helpful to provide details about the training of the facilitators or researchers who delivered the intervention, as this can impact the consistency and effectiveness of the program.
  • You mention several scales (LOT-R, CES-D, ASQ, Grit Scale) but provide limited information about their reliability and validity in your study context. Including this information will strengthen the credibility of your psychometric instruments within your population.
  • Please provide a rationale for selecting specific statistical tests, particularly why MANOVA was chosen for certain comparisons.
  • Lastly, discuss the implications of using a non-probabilistic sampling method on the generalizability of your findings.
  • How do the authors manage measurement bias? I recommend using the Herman Single Factor method solution, along with Variance Inflation Factor (VIF) and Tolerance, to minimize measurement bias. Applying these methods would be beneficial
  1. Discussion and Interpretation:
    • In discussing your results, delve deeper into the implications of your findings. How do they align with or challenge existing literature? What specific aspects of the Hallenges program do you attribute to the observed changes?
    • Address potential confounding variables that could have influenced your results. Discuss how you controlled for these variables or how they might be accounted for in future research.
  2. Limitations and Future Research:
    • Acknowledge any limitations in your study, such as sample diversity or potential biases in self-reported measures. Discuss how these limitations might affect the interpretation of your findings.
    • Suggest concrete next steps for research, such as the potential for longitudinal studies to assess the long-term effects of the Challenges program or adaptations for different cultural contexts.
  3. Conclusion:
    • The conclusion should succinctly summarize the key findings and their implications for practice. Consider emphasizing how your research can inform future interventions in educational settings specifically..

In conclusion, this study has the potential to contribution and significantly contribute to the field and warrants consideration for publication and recommended revision.

Thank you once again for the opportunity to review this work.

References

Hair, J., Black, W., Babin, B., & Anderson, R. (2019). Multivariate Data Analysis: Vol. 8th Editio. Annabel Ainscow.

Author Response

(The authors gave the same response as above.)

Round 2

Reviewer 2 Report

Comments and Suggestions for Authors

Dear Authors,

Congratulations on your outstanding manuscript and the careful revision. I have no additional comments.

With best wishes!